# Influence of Copper Valence in CuO_x_/TiO_2_ Catalysts on the Selectivity of Carbon Dioxide Photocatalytic Reduction Products

**DOI:** 10.3390/nano14231930

**Published:** 2024-11-29

**Authors:** Sha Ni, Wenjing Wu, Zichao Yang, Min Zhang, Jianjun Yang

**Affiliations:** National & Local Joint Engineering Research Center for Applied Technology of Hybrid Nanomaterials, Henan University, Kaifeng 475001, China; 16627557560@163.com (S.N.); wuwenjing@henu.edu.cn (W.W.); 104753231846@henu.edu.cn (Z.Y.); zm1012@henu.edu.cn (M.Z.)

**Keywords:** photocatalytic CO_2_ reduction, TiO_2_, Cu valence state, production selectivity, cocatalyst

## Abstract

The Cu cocatalyst supported on the surface of TiO_2_ photocatalysts has demonstrated unique activity and selectivity in photocatalytic CO_2_ reduction. The valence state of copper significantly influences the catalytic process; however, due to the inherent instability of copper’s valence states, the precise role of different valence states in CO_2_ reduction remains inadequately understood. In this study, CuO_x_/TiO_2_ catalysts were synthesized using an in situ growth reduction method, and we investigated the impact of various valence copper species on CO_2_ photocatalytic reduction. Our results indicate that Cu^+^ and Cu^0^ serve as primary active sites, with the selectivity for CH_4_ and CO products during CO_2_ photoreduction being closely related to their respective ratios on the catalyst surface. The adsorption and activation mechanisms of CO on both Cu^+^ and Cu^0^ surfaces are identified as critical factors determining product selectivity in photocatalytic processes. Furthermore, it is confirmed that Cu^+^ primarily facilitates CH_4_ production while Cu^0^ is responsible for generating CO. This study provides valuable insights into developing highly selective photocatalysts.

## 1. Introduction

With the rapid advancement of global industry and the growth of population, human consumption of fossil energy is escalating on a daily basis. On the one hand, this leads to an energy shortage [1,2]. On the other hand, the substantial emissions of CO_2_ give rise to global warming, glacier melting, the extinction of plankton, and other issues, posing a serious threat to human health and safety [3,4,5]. In the treatment of CO_2_, converting CO_2_ directly into energy substances and chemicals is of the utmost scientific significance. Common strategies for CO_2_ resource utilization include CO_2_ capture and storage [6,7], electrocatalytic CO_2_ reduction [8,9], thermal-assisted catalytic CO_2_ reduction [10,11], and photocatalytic CO_2_ reduction [12,13]. Photocatalytic CO_2_ reduction harnesses the abundant and renewable energy of sunlight to transform CO_2_ into valuable chemicals, all while avoiding the consumption of precious electrical or thermal energy. This process is regarded as an optimal solution to the challenge posed by excessive CO_2_ emissions [14]. Since 1979, it has been reported that TiO_2_ can catalyze the conversion of CO_2_ into CH_3_OH, HCHO, and other chemical compounds, which has sparked a global research surge in photocatalytic CO_2_ conversion to energy-rich substances [14]. Currently, a diverse array of semiconductor photocatalytic materials has been developed, with TiO_2_-based photocatalysts being the most extensively studied and applied in the field of photocatalysis due to their superior efficiency and stability [15,16,17,18,19,20], suitable bandgap structure, low toxicity, and other characteristics [21,22,23]. Nevertheless, the photocatalytic reduction of CO_2_ using TiO_2_ still faces challenges related to limited photocatalytic efficiency and low product selectivity [24]. In order to improve the photocatalytic performance of TiO_2_, surface modification can be carried out by the following common methods: doping [25,26], defect construction [27,28,29], morphology regulation [30], heterojunction construction [31,32], co-catalyst support [33,34], and surface sensitization [35,36]. Among them, supported cocatalyst is an effective means to improve the photocatalytic performance of TiO_2_ [33,34,37,38].

Transition metals are often used as cocatalysts to improve the photocatalytic activity of semiconductors, among which Cu is widely used in the field of photocatalysis, especially in the study of photocatalytic CO_2_ reduction because of its abundant reserves, low price, easy obtainability, and efficient charge separation ability [39,40,41,42,43]. In addition, the use of p-type semiconductors (i.e., copper oxide) in designing catalyst strategies not only enhances charge redistribution due to their narrower band gap but also enhances the selectivity of the reaction to methanol [44]. The remarkable characteristic of Cu is the diversity of valence states (Cu^2+^, Cu^+^ and Cu^0^). Through a series of experiments and characterization methods, it is revealed that in the presence of oxygen, the oxidation of cuprous copper and zero-valent copper affects its performance and stability in the photocatalytic reaction [45,46]. The role of different valence states of Cu in photocatalytic CO_2_ reduction has attracted more and more attention. Many studies have reported the effect of copper on the selectivity of CO_2_ products in photocatalytic reduction [47,48,49,50,51]. Kreft et al. [48] reported a study on the control of different valence Cu components by introducing O_2_ and found that significantly increased product yield and complete selectivity to CO products could be observed in the presence of O_2_, and Cu_2_O was the most active species in the photocatalytic CO_2_ reduction process. When the proportion of Cu_2_O increases, the output of the corresponding product CO will also increase. Zhang et al. [52] reported a catalyst of oxygen-containing copper (Cu_4_O), which showed that CO on the surface of Cu_4_O with oxygen vacancy often continued to hydrogenate to produce high-value products, rather than desorption to produce CO. Different valence Cu components have different adsorption and activation capacities for CO and CO_2_, so the valence of Cu components on the catalyst surface is an important factor affecting CO_2_ reduction. However, due to the instability of the valence state of Cu, there is still a lack of research on the practical role of each valence Cu component in the process of CO_2_ reduction.

The crystalline phases of TiO_2_ are anatase, rutile, plate titanite, and a mixture of anatase and rutile (e.g., P25). Anatase-phase titanium dioxide has a good basis for photocatalytic activity and is easy to be treated and modified. Surface modification and modification can be performed by doping, loading, and other methods to improve its photocatalytic activity and stability [53,54]. More importantly, the crystal structure of anatase-phase titanium dioxide is relatively simple, the property is more stable, and it is easy to characterize and analyze. In this study, the surface of anatase TiO_2_ was modified with different valence Cu species. The CuO_x_/TiO_2_ catalyst was prepared with an in situ growth reduction method to explore the effects of different valence Cu components on the photocatalytic reduction of CO_2_. Combined with a catalyst characterization test and photocatalytic CO_2_ reduction performance test, the results showed that the introduction of CuO_x_ did not change the structure, morphology, or redox potential of TiO_2_. Cu^+^ and Cu^0^ are the main active sites on the catalysts. The selectivity of CH_4_ and CO in the photocatalytic reduction of CO_2_ products by CuO_x_/TiO_2_ is related to the ratio of Cu^+^ and Cu^0^ content on the catalyst surface. The mechanism of the influence of different valence copper components on product selectivity was analyzed from the perspective of thermodynamics and kinetics. *CO is more likely to experience desorption from the Cu^0^ surface to produce CO while continuing to adsorb on the Cu^+^ surface to produce CH_4_. In this study, the adsorption of CO on different valence Cu components and the relationship between Cu valence and corresponding products in the process of CO_2_ photocatalytic reduction were investigated, which provided a guide for the development of highly selective photocatalysts.

## 2. Experimental

### 2.1. Chemical Materials

The chemicals used in the experiment were purchased from commercial suppliers without further treatment. They are titanium dioxide (anatase, 99.8%, Aladdin, Shanghai, China), copper(II) acetate monohydrate (C_4_H_6_CuO_4_·H_2_O, ≥98.0%, Kermel, Colmar, France), ethylene glycol (EG, (CH_2_OH)_2_, AR), and ethanol (CH_3_CH_2_OH, AR).

### 2.2. Synthesis of the CuO_x_/TiO_2_

An amount of 500 mg TiO_2_ and 32.14 mg Cu(CH_3_COO)_2_·H_2_O were dispersed into a mixture of 30 mL H_2_O and ethylene glycol, stirred at room temperature for 1 h, and then hydrothermal reaction was carried out in an oven at 200 °C for 2 h, cooled to room temperature, washed with H_2_O and ethanol three times, respectively, and dried at 60 °C for 12 h. Solid powder catalyst was obtained by grinding. According to the addition of y mL ethylene glycol (reductant), the catalyst sample was denoted as CuO_x_/TiO_2_-y (y = 0 ~ 5).

### 2.3. Characterization

The crystal structures of CuO_x_/TiO_2_ were characterized using X-ray diffraction (XRD, D8-ADVANCE, Cu Kα radiation, 2θ = 20 ~ 80°, China), whose operation voltage and current were set at 40 kV and 30 mA, respectively; the morphologies were analyzed by transmission electron microscopy and high-resolution transmission electron microscopy equipped with FFT; (TEM, HRTEM, JEOL JEM-F200, Tokyo, Japan). The UV–Vis diffuse reflectance spectra (DRS) absorbance spectra were obtained with a Scan UV–Vis diffuse reflectance spectrophotometer (Shimadzu, Kyoto, Japan, UV-2600), using BaSO_4_ as the reflectance sample. Specific surface area, pore size distribution, and CO_2_ physical adsorption spectra were measured by BET surface area measurements (Quadrascorb SI-4), which were carried out by N_2_ adsorption–desorption isotherms. CO_2_/TPD was carried out on the ChemBET PULSARTMTPR/TPD chemisorption analyzer with argon (Ar) as the carrier gas. The surface chemical states of elements on different samples were characterized by X-ray photoelectron spectroscopy and Auger electron spectroscopy (XPS and AES, Thermo ESCALAB 250 Xi, Waltham, MA, USA), and the shift of the spectra is due to the relative surface charging corrected according to the standard binding energy of C 1 s at 284.6 eV. The oxygen vacancy was measured by electron paramagnetic resonance (EPR, Bruker A300-10/12, Billerica, MA, USA). The real loading amount of Cu in samples was measured by an inductively coupled plasma optical emission spectrometer (ICP-OES, Agilent ICPOES730, Santa Clara, CA, USA). The electrochemical measurement was performed on an electrochemical analyzer (CHI600E) with three electrode cells at room temperature. The working electrodes were made of ITO glass and the corresponding prepared samples. The Na_2_SO_4_ aqueous solution (0.1 M) was used as an electrolyte and a 300 W Xe lamp (PLS-SXE300/300UV) was used as the light source. Photoluminescence spectra (PL) were recorded on a confocal laser Raman microscope (HORIBA FLuoroMax+, Kyoto, Japan) using a 310 nm excitation light source; the time-resolved transient PL decay of samples was measured by transient fluorescence spectrometer (Edinburgh FLS 980, Edinburgh, UK).

The in situ FT-IR was tested on a Bruker Tensor II spectrometer (Billerica, MA, USA). The samples were loaded into the in situ reaction tank of the infrared spectrometer, and the sample was pretreated for 1 h under vacuum at 80 °C. The photocatalytic reaction process was simulated: CO_2_ and 0.5 mL H_2_O were slowly injected, the adsorption–desorption equilibrium was reached after adsorption for 1 h, and the reaction system was illuminated. The infrared spectra under different illumination times were collected and the process of CO_2_ reduction catalyzed by catalysts was analyzed. FT-IR spectra of adsorbed CO were measured using the same pretreatment method. After pretreatment, CO was slowly injected, adsorbed for 1 h, and then excess CO was removed. The temperature of all samples gradually increased from 30 °C to 50 °C and was measured after being stabilized at each temperature for 5 min.

### 2.4. Photocatalytic Reduction Reaction of CO_2_

The performance of photocatalytic CO_2_ reduction was tested in a custom-made quartz glass reactor with a volume of 0.3 L. The 20 mg catalyst powder sample was laid on the surface of the small circular table at the bottom, and 1 mL deionized water was uniformly added around the circular table. After sealing with a quartz lid, CO_2_ gas was poured into the reactor for 1 h to empty it, and the reactor was filled with CO_2_ at the same time. After ventilation, both ends of the reactor were sealed. A 300 W xenon lamp (PLS-SXE300/300UV, China) was used as the lamp source for analysis by gas chromatograph (Shimadzu, GC-2018). Xenon lamps were illuminated from top to bottom through the quartz glass cover of the reactor, and 1 mL of the gas in the reactor was collected every 1 h and injected into the gas chromatograph for quantitative detection of CO and CH_4_. The retention time and standard curve of each component gas were obtained by detecting the standard gas.

## 3. Results and Discussion

### 3.1. The Structure and Morphology

The crystal structure of the catalyst was analyzed using X-ray diffraction (XRD) and Raman spectroscopy. As shown in Figure 1a, XRD patterns of CuOx/TiO_2_ show typical anatase-phase TiO_2_ (JCPDS No. 84-1286) crystal structure, indicating that the introduction of copper does not change the crystal structure of TiO_2_. The diffraction peaks observed at 2θ = 36.1° and 61.4° correspond to the (111) and (220) crystal faces of Cu_2_O (JCPDS No. 78-2076), respectively. The diffraction peaks observed at 2θ = 43.3° and 50.4° correspond to the (111) and (200) crystal faces of Cu (JCPDS No. 85-1326), respectively, indicating that copper species are successfully supported on the TiO_2_ surface in the form of Cu_2_O or Cu. As shown in Figure 1b, Raman diagrams of different catalyst samples all show the Raman characteristic peaks of anatase-phase TiO_2_, and the peak position of anatase-phase TiO_2_ does not shift after the introduction of copper, further indicating that the crystal phase of anatase-phase TiO_2_ remains unchanged before and after the reaction.

The surface morphology of the catalyst was analyzed by transmission electron microscopy (TEM). The results reveal that, as shown in Figure 2a,b, there is no significant difference in the morphology of CuO_x_/TiO_2_-2 and TiO_2_, and the size is about 50 nm nano-sheets, indicating that the introduction of copper has no effect on the morphology of TiO_2_. This is further confirmed by BET surface area measurements. As shown in the figure, TiO_2_, CuO_x_/TiO_2_-2, and CuO_x_/TiO_2_-5 have no obvious differences in specific surface area. The results show that the addition of Cu cannot change the surface structure of TiO_2_, which is consistent with the results of TEM. The HRTEM image of CuO_x_/TiO_2_-2 is shown in Figure 2c. The lattice fringes with a lattice spacing of 0.246 nm, 0.208 nm, and 0.351 nm correspond to the Cu_2_O (111) crystal face, Cu (111) crystal face, and anatase-phase TiO_2_ (101) crystal face, respectively. This is consistent with the XRD results. In Figure 2d–f, element distribution mapping shows the uniform distribution of Ti, O, and Cu elements. The loading capacity of Cu species on CuO_x_/TiO_2_-2 was 1.16 wt.% by ICP-OES. The above results show that Cu species are uniformly supported on the surface of TiO_2_ in the form of Cu0 and Cu_2_O, and are closely bound to TiO_2_.

### 3.2. Copper Valence State on the Surface of CuO_x_/TiO_2_

In the process of the photocatalytic reduction of CO_2_ by CuO_x_/TiO_2_, Cu species are the active center of the catalytic reduction reaction, and the valence of Cu plays an important role in the selectivity of CO_2_ products by photocatalytic reduction. According to the XRD pattern in Figure 1a, it can be preliminarily concluded that the present state of Cu changes regularly with the additional amount of glycol as a reducing agent. The valence state of Cu in the catalyst was further analyzed qualitatively and quantitatively with XPS and Auger electron spectroscopy (AES). Firstly, XPS was used to characterize the peak position of each valence Cu component on the catalyst surface for qualitative analysis, and the proportion of the peak area of each valence Cu component to the total peak area of all Cu components was calculated for quantitative analysis. As shown in Appendix A, in the XPS spectrum of Cu 2p, the characteristic peaks at the binding energies of 932.08 eV and 952.08 eV correspond to the Cu 2p_3/2_ and Cu 2p_1/2_ orbitals of Cu^+^/Cu^0^, respectively, occupying the majority of the peak area of copper components. Its proportion increased slightly with the increase in the amount of ethylene glycol. The characteristic peaks at 934 eV and 935 eV correspond to the Cu 2p_3/2_ and Cu 2p_1/2_ orbitals of Cu^2+^, respectively, and occupy a small part of the peak area of the copper component, which decreases slightly with the increase in the amount of ethylene glycol added. Appendix A shows that the Cu component mainly exists in the form of Cu^+^/Cu^0^, but the two components cannot be distinguished by XPS and need to be further analyzed with Auger electron spectroscopy (AES).

The AES energy spectrum of Cu LMM is shown in Figure 3a–e. The characteristic peak at 569.6 eV binding energy corresponds to Cu^+^, the characteristic peak at 567.6 eV binding energy corresponds to Cu^0^, and the characteristic peak at 564.5 eV is related to the Ti 2s orbital. Figure 3f shows that the proportion of Cu^0^ on the catalyst surface increases with the addition of ethylene glycol, while the proportion of Cu^+^ decreases. Due to the fact that the catalyst cannot avoid contact with oxygen in the air during the test, and the copper valence state is unstable, the test results show that the copper component cannot exist completely in the form of Cu^0^.

### 3.3. Performance of Photocatalytic CO_2_ Reduction

The performance of the catalyst for photocatalytic CO_2_ reduction was tested. The yields and selectivity of CO_2_ products for different samples are shown in Figure 4a. The photocatalytic CO_2_ reduction performance of pure TiO_2_ was poor, with a CO yield of 1.41 μmol·g^−1^·h^−1^ and CH_4_ yield of 0.28 μmol·g^−1^·h^−1^. The photocatalytic CO_2_ reduction performance of TiO_2_ was improved after loading CuO_x_. Comparing the photocatalytic activities of different samples, it can be found that with the increase of reduction degree, the Cu valence state decreases, and the CO yield gradually increases. The CO yield of CuO_x_/TiO_2_-5 is up to 10.68 μmol·g^−1^·h^−1^, and the CO selectivity is up to 80.12%. The CH_4_ yield of CuO_x_/TiO_2_-2 was up to 10.8 μmol·g^−1^·h^−1^, and the selectivity of CH_4_ was up to 71.9%. Comparing the selectivity of CO_2_ products by photocatalytic reduction of different samples, it can be found that Cu^2+^ has little effect on the selectivity of products. With the gradual reduction of Cu^2+^, the selectivity of CO first decreases and then gradually increases, and the selectivity of CH_4_ first increases and then decreases. According to the analysis of the proportion of copper content in different valence states in Figure 4b, it can be seen that copper species Cu^+^ and Cu^0^ help to improve the reduction ability of the catalyst. With the increase of Cu^0^ content, the selectivity of CO gradually increases, indicating that Cu^0^ is conducive to the selective conversion of CO, while Cu^+^ is conducive to the selective conversion of CH_4_. These results indicate that the valence state of Cu may be the key to the selectivity of CO_2_ products in photocatalytic reduction. As shown in Table 1, CuO_x_/TiO_2_ has higher photocatalytic activity than other photocatalysts.

In the process of photocatalytic CO_2_ reduction, considering that many factors have certain effects on the activity and selectivity of the reaction, we tested the photocatalytic CO_2_ reduction activity of CuO_x_/TiO_2_-2 under different reaction conditions. As shown in Appendix A, a small amount of CO and CH_4_ were produced in the Ar atmosphere, which was generated by a small amount of carbon-containing reagents remaining in the catalyst preparation process. Compared with the results of the reaction activity and product selectivity in the CO_2_ atmosphere, the results were negligible, indicating that the carbon source of the photocatalytic reduction reaction products mainly came from CO_2_ gas. In the dark state, no CO_2_ reduction products were detected in the reaction system of the catalyst, indicating that light is the necessary condition for the CuO_x_/TiO_2_ photocatalytic reduction of CO_2_ reaction.

### 3.4. The Selectivity of Products in Photocatalytic Reduction of CO_2_

The adsorption of CO_2_ by a catalyst is the first step of the photocatalytic CO_2_ reduction reaction. The physical and chemical adsorption capacity of different samples for CO_2_ is analyzed by using a specific surface area and an aperture analyzer and chemical absorption desorption instrument (CO_2_/TPD). As shown in Figure 5a, the physical adsorption capacity of the catalyst remained basically unchanged after the introduction of copper, which was consistent with the test results of the N_2_ resorption–desorption curve. In Figure 5b CO_2_/TPD, the test results show that the CO_2_/TPD curve of TiO_2_ shows the desorption of CO_2_ at low temperatures, and the corresponding temperatures of desorption peaks are 101 °C and 322 °C, respectively, indicating that the interaction between CO_2_ and TiO_2_ is relatively low. When the desorption temperatures of CuO_x_/TiO_2_-2 and CuO_x_/TiO_2_-5 are higher, the desorption temperatures are 356 °C, 500 °C, 345 °C, and 475 °C respectively, indicating that the interaction between CO_2_ and CuO_x_/TiO_2_-2 and CuO_x_/TiO_2_-5 is stronger. The results show that CuO_x_ cocatalyst can enhance the interaction between CO_2_ and photocatalyst. CuO_x_/TiO_2_-2 showed a higher desorption peak than CuO_x_/TiO_2_-5, indicating that Cu^+^ is more conducive to the chemical adsorption of CO_2_ and the subsequent photocatalytic reduction of the CO_2_ reaction process.

The UV–Vis DRS spectra are shown in Figure 5c. Compared with pure TiO_2_, CuO_x_/TiO_2_ had visible light absorption, indicating that the introduction of copper increased the light absorption range of the catalyst. In addition, the absorption band edge of the catalyst remains basically unchanged (387 nm), indicating that the supported Cu does not change the band gap of TiO_2_, which is consistent with the results of XRD and Raman. As shown in Figure 5d, CuO_x_/TiO_2_ has a higher photocurrent response than pure TiO_2_, among which CuO_x_/TiO_2_-2 has the strongest photocurrent response, indicating that the introduction of copper improves the transport capacity of photogenerated electrons and increases the mobility of electrons. However, the selectivity of the photocatalytic reduction of CO_2_ products cannot be directly determined by electron migration. Therefore, steady-state fluorescence spectra and fluorescence lifetime test were used to further analyze the lifetime of photogenerated electrons. As shown in Figure 5e,f, the catalyst has strong characteristic peaks at wavelengths of 394 nm and 466 nm, and the fluorescence intensity decreases after the introduction of copper, which indicates that the presence of Cu can promote the migration of photogenerated electrons to the surface of the catalyst, inhibiting the recombination of photogenerated electrons and holes. The fluorescence lifetime of TiO_2_ is 6.00 ns, the fluorescence lifetime of CuO_x_/TiO_2_-2 is 6.40 ns, and the fluorescence lifetime of CuO_x_/TiO_2_-5 is 6.21ns. The introduction of copper can migrate electrons from TiO_2_ to CuO_x_, extending the electron lifetime. From the perspective of reaction kinetics, since the generation of CH_4_ is an eight-electron reaction, the catalyst with a long photo-generated electron lifetime is more likely to generate CH_4_ products in the photocatalytic reduction of CO_2_, which also explains the higher selectivity of CuO_x_/TiO_2_-2 photocatalytic reduction of CO_2_ to CH_4_.

According to the semiconductor band gap (E_g_) formula, (αhν)^n^ = k(hν − Eg), the Tauc of different samples is calculated, as shown in Figure 6a. The band gap of TiO_2_ is about 3.2 eV, and the band gap remains basically unchanged after loading CuO_x_. VB-XPS was used to directly test the valence band position of the catalyst, and the results are shown in Figure 6b. The valence band values of different samples under standard hydrogen electrodes were calculated according to the following formula: E_VB,NHE_ = φ + E_VB,XPS_ − 4.44, where φ is the work function of the instrument (4.5 eV). Therefore, the E_VB, NHE_ of CuO_x_/TiO_2_-2 and CuO_x_/TiO_2_-5 are calculated to be 2.42 eV and 3.32 eV, respectively. According to the formula E_VB_ = E_CB_ + E_g_, the conduction bands (E_CB_) of CuO_x_/TiO_2_-2 and CuO_x_/TiO_2_-5 are −0.75 eV and −0.83 eV, respectively. As shown in Figure 6c, the band conduction position of TiO_2_ was calculated to be −0.41 eV. Based on the above results, the band gap relationship of different samples is shown in Figure 6d. The conduction position of CuO_x_/TiO_2_ is more negative than that of single TiO_2_, indicating that loaded CuO_x_ can improve the photocatalytic reduction ability of TiO_2_ and enhance the photocatalytic reduction activity of CO_2_. The conduction positions of CuO_x_/TiO_2_-2 and CuO_x_/TiO_2_-5 are more negative than the reaction potentials of CH_4_ (CH_4_/CO_2_, −0.24 V vs. NHE) and CO (CO/CO_2_, −0.52 V vs. NHE). The results show that the photocatalytic reduction of CO_2_ to produce CH_4_ and CO is thermodynamically feasible. Combined with the UV–Vis DRS results, the effect of different valence states of Cu on the redox potential of CuO_x_/TiO_2_ catalyst is not significant. In other words, under these conditions, the changes in product yield and the selectivity of CuO_x_/TiO_2_ photocatalytic reduction of CO_2_ are not determined by the redox potential of the catalyst.

Carbon monoxide (CO) is not only a significant product of photocatalytic CO_2_ reduction but also serves as an important reaction intermediate. The key to producing high-value products lies in the continued adsorption of CO on the catalyst surface, preventing its desorption during the photocatalytic CO_2_ reduction process. Fourier Transform Infrared Spectroscopy (FT-IR) was employed to assess the adsorption capacity of various samples for CO. A stronger interaction between the catalyst and CO correlates with increased difficulty in desorption from the catalyst surface. As temperature rises, the rate of decrease in the *CO signal intensity diminishes. The reduction rates of the *CO characteristic peak intensities at 2171 cm^−1^ and 2100 cm^−1^ can be utilized to characterize the CO adsorption capacity of the photocatalyst [60,61,62].

As shown in Figure 7a, when the temperature rises to 30 °C, the *CO signal on the pure TiO_2_ sample begins to decline rapidly, and when the temperature rises to 45 °C, the *CO adsorption peak completely disappears, indicating that the interaction force between CO and TiO_2_ is weak. Compared with a pure TiO_2_ sample, the *CO absorption peak of the catalyst CuO_x_/TiO_2_ decreased at a slower rate, indicating that the interaction force between CO and CuO_x_/TiO_2_ was strong, that is, the main adsorption site of CO was CuO_x_. The FT-IR spectra of CO adsorption on CuO_x_/TiO_2_-2 and CuO_x_/TiO_2_-5 are shown in Figure 7b,c, respectively. The effects of different valence states of Cu on CO adsorption are compared and analyzed. The results show that the decline rate of *CO signal intensity is as follows: R_TiO2_ > R_CuOx/TiO2-5_ > R_CuOx/TiO2-2_, so the adsorption capacity of CO is CuO_x_/TiO_2_-2 > CuO_x_/TiO_2_-5 > TiO_2_. This is consistent with the change in the ratio of copper components in the AES spectra. With the decrease of the ratio of Cu^+^ and the increase of the ratio of Cu^0^, the adsorption capacity of CO decreases, and CO is more easily resolved to form CO products. This shows that the adsorption capacity of Cu^+^ for CO is stronger than that of Cu^0^, which is conducive to further hydrogenation of CO, and the final product CH_4_ is formed through a carbene pathway.

As shown in Figure 8a, the adsorption peaks of HCO_3_^−^ (1415 cm^−1^), m-CO_3_^2−^ (1506 and 1447 cm^−1^), and b-CO_3_^2−^ (1576 and 1522 cm^−1^) can be observed in the in situ FT-IR spectra of CuO_x_/TiO_2_-2. The intensity of these characteristic adsorption peaks increased gradually with the extension of adsorption time, but the location did not change. COOH* (1558 cm^−1^), *CHO (1102 cm^−1^), and CH_3_O* (1041 cm^−1^) peaks appeared and increased with the increase of light time. They are all important intermediate species in the process of photocatalytic CO_2_ reduction. In addition, CH_3_O*, *CHO, and CH_2_· (1373 cm^−1^) [41] participate in the reaction as important intermediates of CH_4_, which explains the high selectivity of the CuO_x_/TiO_2_-2 catalyst for CH_4_ generation. It is speculated that the main conversion pathway of CH_4_ is the carbene pathway: CO_2_ → *COOH → *CO → *CHO → C· → ·CH_2_ → ·CH_3_ → CH_4_. As shown in Figure 8b, in the photocatalytic CO_2_ reduction process of CuO_x_/TiO_2_-5, the intermediates are mainly CO_3_^2−^, HCO_3_^−^, and *COOH, and no absorption peak of methane intermediates is observed. This indicates that *CO does not accumulate on the catalyst surface for further conversion but is quickly released into the air and converted into the final product CO. This is consistent with the results of the CO adsorption FT-IR, which explains the high selectivity of the CuO_x_/TiO_2_-5 photocatalytic reduction of CO_2_ to produce CO. It is speculated that the main conversion pathway of CO is as follows: CO_2_ → HCO_3_^−^/CO_3_^2−^ → *COOH → CO. Different catalysts produce different reaction intermediates in the light process, which directly affect the yield and selectivity of the final photocatalytic reduction of CO_2_ products.

Building on the aforementioned results, a reaction mechanism for the CuO_x_/TiO_2_ photocatalytic reduction of CO_2_ has been proposed. The CuO_x_ species supported on the TiO_2_ surface serve as a reactive site that effectively harnesses photoelectron generation, enhances charge separation efficiency, and boosts the photocatalytic activity for CO_2_ reduction. The results from Auger electron spectroscopy (AES) and evaluations of CO_2_ photoreduction performance indicate that the selectivity towards methane (CH_4_) and carbon monoxide (CO) in the products is closely linked to the relative content ratio of Cu^+^ and Cu^0^ present on the catalyst surface. Furthermore, it was observed that the adsorption and activation of *CO intermediates on this surface significantly influence final product formation. CO readily desorbs from Cu0 to yield CO, while CH_4_ undergoes further adsorption and hydrogenation at Cu^+^, thereby confirming that Cu^+^ acts as an active site for CH_4_ production, whereas Cu^0^ functions as an active site for CO generation (Figure 9).

## 4. Conclusions

The CuO_x_/TiO_2_ photocatalyst was synthesized via an in situ growth reduction method, enabling selective regulation of CO_2_ photoreduction products by modulating the valence state of copper. The valence state of copper is a critical determinant influencing the selectivity of CO_2_ reduction products. Cu^+^ serves as the active site for methane (CH_4_) formation, while Cu^0^ acts as the active site for carbon monoxide (CO) production. Notably, CO is not only a significant product of CO_2_ photoreduction but also functions as an essential reaction intermediate. Cu^+^ exhibits strong adsorption and activation capabilities towards CO, thereby facilitating the conversion of *CO intermediates into high-value CH_4_ products. This study offers valuable insights for advancing research on copper-based photocatalysts and identifying highly efficient and selective catalysts for CO_2_ reduction.

## Figures and Tables

**Figure 1 nanomaterials-14-01930-f001:**
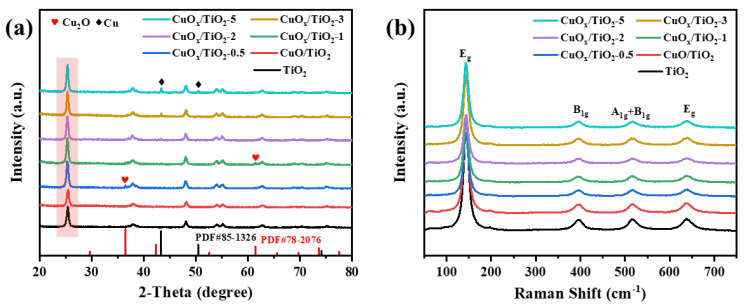
(**a**) XRD patterns and (**b**) Raman of CuO_x_/TiO_2_-y (y = 0.5, 1, 2, 3, 5).

**Figure 2 nanomaterials-14-01930-f002:**
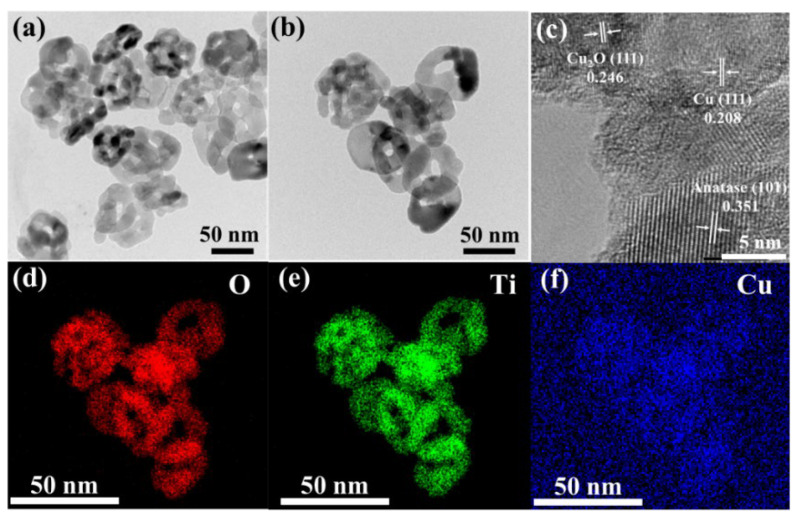
(**a**) TEM diagram of TiO_2_, (**b**) TEM diagram, (**c**) HRTEM diagram, and (**d**–**f**) mapping diagram of CuOx/TiO_2_-2.

**Figure 3 nanomaterials-14-01930-f003:**
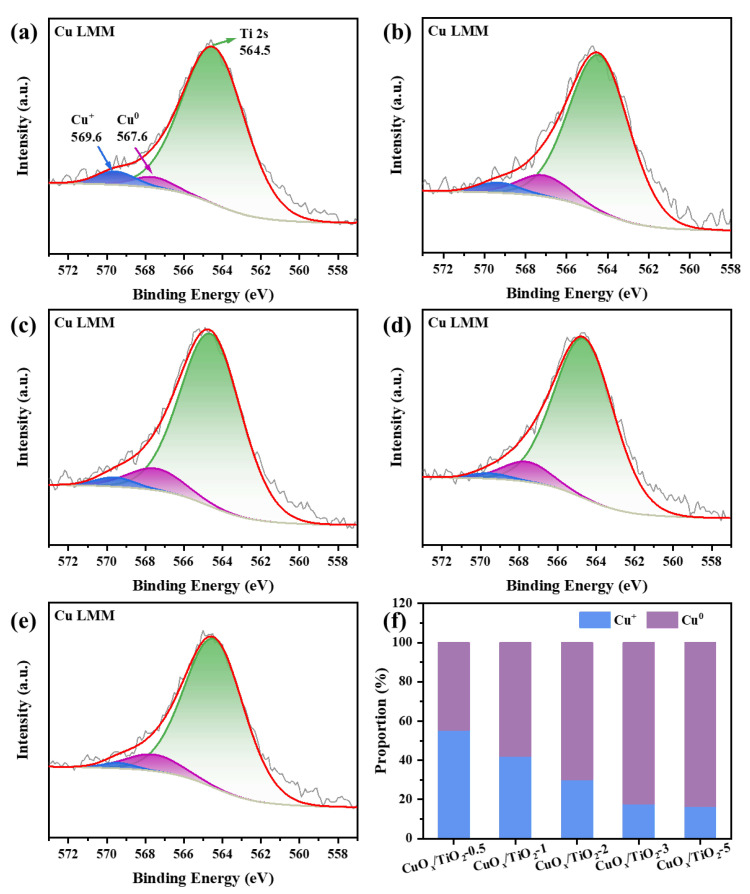
Cu LMM Auger spectra of (**a**–**e**) CuO_x_/TiO_2_-y (y = 0.5, 1, 2, 3, 5); (**f**) proportion of different states of Cu^+^ and Cu^0^ components in CuO_x_/TiO_2_.

**Figure 4 nanomaterials-14-01930-f004:**
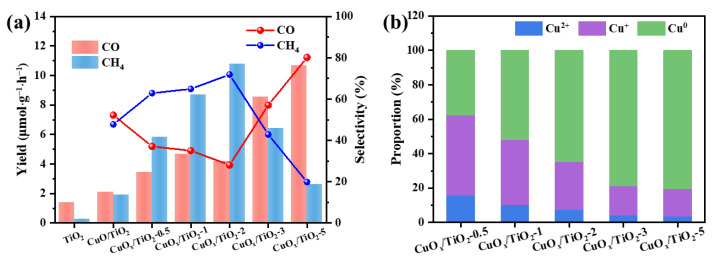
(**a**) Photocatalytic CO_2_ reduction activity of CuO_x_/TiO_2_ and (**b**) proportion of different states of Cu components in CuOx/TiO_2_.

**Figure 5 nanomaterials-14-01930-f005:**
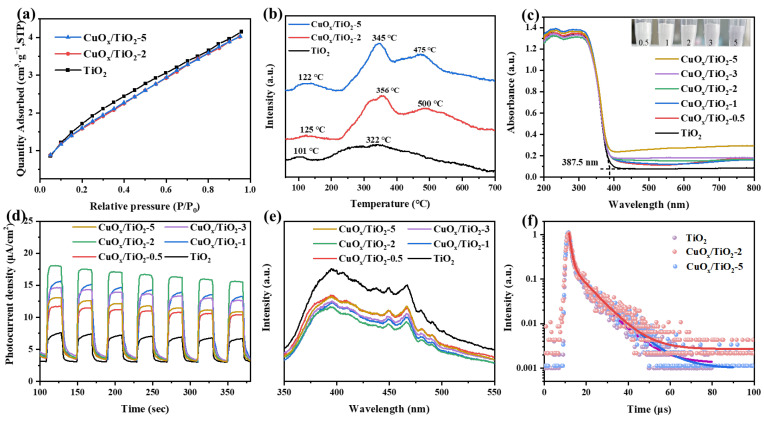
(**a**) CO_2_ physical adsorption image, (**b**) CO_2_ chemisorption (CO_2_/TPD) image, (**c**) UV–Vis DRS image, (**d**) I–t curve, (**e**) steady-state fluorescence spectrum image, and (**f**) transient fluorescence lifetime image of CuO_x_/TiO_2_.

**Figure 6 nanomaterials-14-01930-f006:**
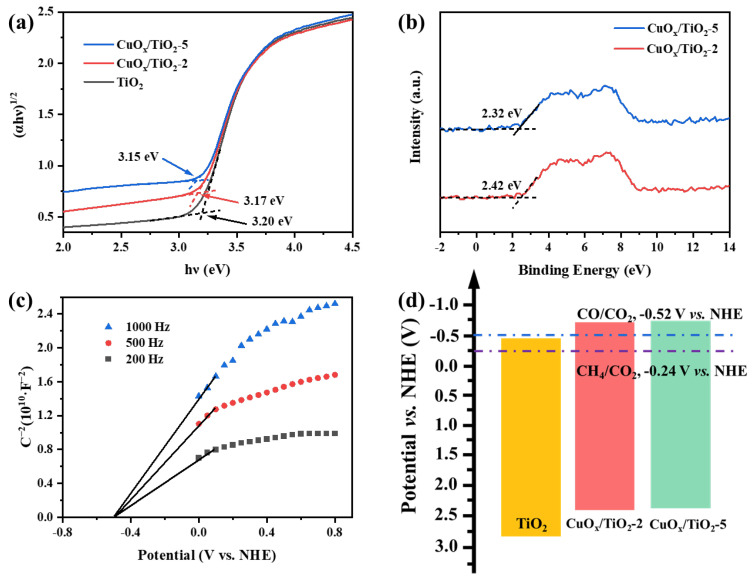
(**a**) Tauc, (**b**) VB-XPS spectrum, (**c**) Mott–Schottky curve of TiO_2_ and (**d**) band structure of catalyst.

**Figure 7 nanomaterials-14-01930-f007:**
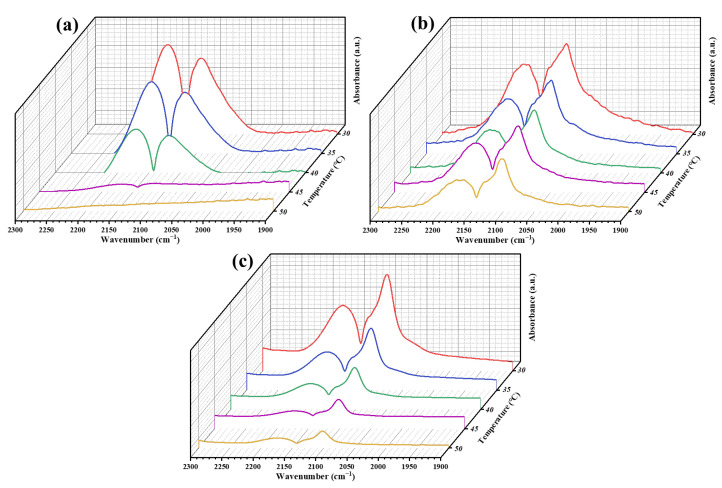
FT-IR spectra of CO adsorbed on (**a**) TiO_2_, (**b**) CuO_x_/TiO_2_-2, and (**c**) CuO_x_/TiO_2_-5.

**Figure 8 nanomaterials-14-01930-f008:**
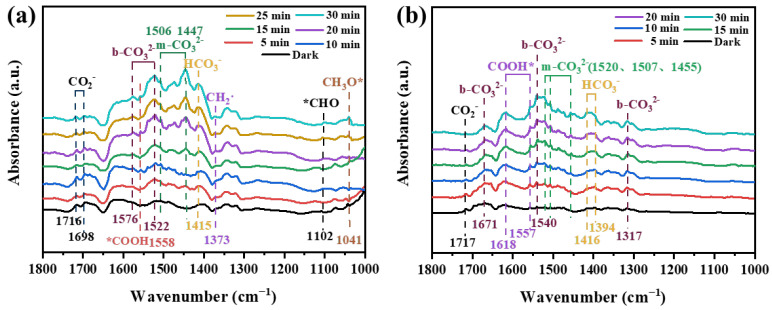
In situ FT-IR spectra of (**a**) CuO_x_/TiO_2_-2 and (**b**) CuO_x_/TiO_2_-5.

**Figure 9 nanomaterials-14-01930-f009:**
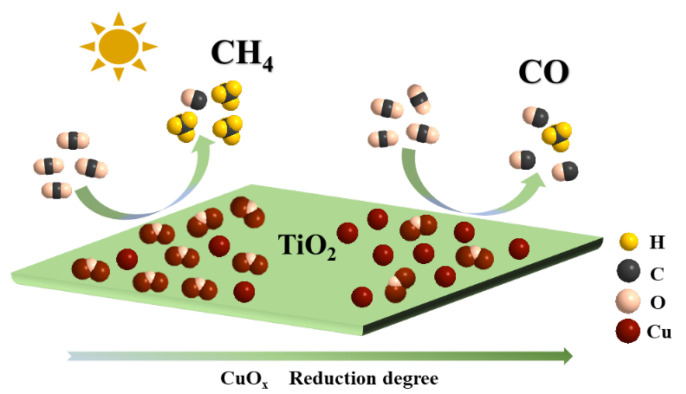
Schematic image of CuO_x_/TiO_2_ photocatalytic reduction of CO_2_.

**Table 1 nanomaterials-14-01930-t001:** Performance comparison between CuO_x_/TiO_2_ and other photocatalysts.

Catalyst Name	Intended Product	Productivity (µmol g^−1^ h^−1^)	Ref.
CuO_x_/TiO_2_	COCH_4_	10.6810.8	This work
TiO_2_/Cu_2_O	CO	10.22	[55]
TiO_2_/Cu_2_O	CH_4_	1.35	[56]
Cu:TiO_2_-CS	COCH_4_	4.485.34	[57]
Cu_2_O/S-TiO_2_/CuO	CH_4_	2.31	[58]
NH_2_–B–TiO_2_–CuO	COCH_4_	3.836.6	[59]

## Data Availability

Data are contained within the article or Appendix A.

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
