# Peer review of "Influence of Copper Valence in CuOx/TiO2 Catalysts on the Selectivity of Carbon Dioxide Photocatalytic Reduction Products"

_nanomaterials, 2024, doi:10.3390/nano14231930_

Round 1
Reviewer 1 Report
Comments and Suggestions for Authors
The photo-assisted conversion of COâ‚‚ into useful products is a research topic that deserves significant attention nowadays. In this report, the authors evaluate how the valence state of Cu controls COâ‚‚ photoreduction selectivity towards CHâ‚„ and CO, demonstrating enhanced performance compared to TiOâ‚‚. The operating principles of the analysis are clear, and technically speaking, the report is valuable and will inspire researchers in the field. Thus, I believe the manuscript might be published in Nanomaterials after addressing the following points:
- The authors may consider discussing recent reports on the use of Cu and Cu/TiOâ‚‚-based surfaces for the photo-assisted reduction of COâ‚‚, where various aspects, including the valence state and reaction selectivity, are evaluated (i.e. Journal of CO2 Utilization, 67, 2023, 102340; ACS Sustainable Chemistry & Engineering, 11, 36. 2023). I believe the discussion can be enhanced by considering these recent reports. It may also provide a more comprehensive literature review.
- It would also be beneficial to include a table or figure comparing the performance achieved for CHâ‚„ and CO with the developed materials to other Cu-based photocatalytic systems reported in the literature. The comparison can be made in terms of productivity and AQY.
Author Response
The photo-assisted conversion of COâ‚‚ into useful products is a research topic that deserves significant attention nowadays. In this report, the authors evaluate how the valence state of Cu controls COâ‚‚ photoreduction selectivity towards CHâ‚„ and CO, demonstrating enhanced performance compared to TiOâ‚‚. The operating principles of the analysis are clear, and technically speaking, the report is valuable and will inspire researchers in the field. Thus, I believe the manuscript might be published in Nanomaterials after addressing the following points:
1. The authors may consider discussing recent reports on the use of Cu and Cu/TiOâ‚‚-based surfaces for the photo-assisted reduction of COâ‚‚, where various aspects, including the valence state and reaction selectivity, are evaluated (i.e. Journal of CO2 Utilization, 67, 2023, 102340; ACS Sustainable Chemistry & Engineering, 11, 36. 2023). I believe the discussion can be enhanced by considering these recent reports. It may also provide a more comprehensive literature review.
- Resonse:
Thank you for your valuable suggestion, some references have been added in the revised manuscript..
Transition metals are often used as cocatalysts to improve the photocatalytic activity of semiconductors, among which Cu is widely used in the field of photocatalysis, especially in the study of photocatalytic CO2 reduction, because of its abundant reserves, cheap and easy to obtain, and efficient charge separation ability. In addition, the use of p-type semiconductors (i.e., copper oxide) in designing catalyst strategies not only enhances charge redistribution due to their narrower band gap, but also enhances the selectivity of the reaction to methanol[1].
2. It would also be beneficial to include a table or figure comparing the performance achieved for CHâ‚„ and CO with the developed materials to other Cu-based photocatalytic systems reported in the literature. The comparison can be made in terms of productivity and AQY. - Response : Thank you for your suggestion. A table was supplemented as follows.
Catalyst Name
Intended Product Productivity (µmol g−1 h−1)
Ref. CuOx/TiO2
CO
CH4
10.68
10.8
This work
TiO2/Cu2O
CO
10.22
[4]
TiO2/Cu2O
CH4
1.35
[5]
Cu:TiO2-CS
CO
CH4
4.48
5.34
[6]
Cu2O/S-TiO2/CuO
CH4
2.31
[7]
NH2–B–TiO2–CuO
CO
CH4
3.83
6.6
[8]
Table 1 Performance comparison between CuOx/TiO2 and other photocatalysts. - Rreferences
[1] Merino-Garcia I, García G, Hernández I, Albo J. An optofluidic planar microreactor with photoactive Cu2O/Mo2C/TiO2 heterostructures for enhanced visible light-driven CO2 conversion to methanol [J]. Journal of CO2 Utilization, 2023, 67: 102340.
[2] Mu X, Wang K, Lv K, et al. Doping of Cr to Regulate the Valence State of Cu and Co Contributes to Efficient Water Splitting [J]. ACS Applied Materials & Interfaces, 2023, 15(13): 16552-61.
[3] Sun T, Gao F, Wang Y, et al. Morphology and valence state evolution of Cu: Unraveling the impact on nitric oxide electroreduction [J]. Journal of Energy Chemistry, 2024, 91: 276-86.
[4] Qian H, Yuan B, Liu Y, et al. Oxygen vacancy enhanced photocatalytic activity of Cu2O/TiO2 heterojunction [J]. iScience, 2024, 27(5).
[5] Yang G, Qiu P, Xiong J, et al. Facilely anchoring Cu2O nanoparticles on mesoporous TiO2 nanorods for enhanced photocatalytic CO2 reduction through efficient charge transfer [J]. Chinese Chemical Letters, 2022, 33(8): 3709-12.
[6] She H, Zhao Z, Bai W, et al. Enhanced performance of photocatalytic CO2 reduction via synergistic effect between chitosan and Cu:TiO2 [J]. Materials Research Bulletin, 2020, 124: 110758.
[7] Kim H R, Razzaq A, Grimes C A, In S-I. Heterojunction p-n-p Cu2O/S-TiO2/CuO: Synthesis and application to photocatalytic conversion of CO2 to methane [J]. Journal of CO2 Utilization, 2017, 20: 91-6.
[8] Chen Z, Li L, Cheng G. Selectively anchoring Cu(OH)2 and CuO on amine-modified brookite TiO2 for enhanced CO2 photoreduction [J]. Carbon Letters, 2023, 33(5): 1395-406.
[9] Zhao Y, Shu Y, Linghu X, et al. Modification engineering of TiO2-based nanoheterojunction photocatalysts [J]. Chemosphere, 2024, 346: 140595.
[10] Rafique M, Hajra S, Irshad M, et al. Hydrogen Production Using TiO2-Based Photocatalysts: A Comprehensive Review [J]. ACS Omega, 2023, 8(29): 25640-8.
Reviewer 2 Report
Comments and Suggestions for Authors
The authors report the preparation of heterostructured photocatalysts associating anatase TiO2 and Cu(0) or copper oxides. The key role played by the oxidation state of Cu in the photocatalytic CO2 reduction was demonstrated. The work is of interest and the catalysts were well characterized. From my opinion, results could be better discussed in the context of literature. The following comments should be considered:
- introduction : "instability of Cu". Please clarify. Only Cu(0) and Cu+ are unstable in the presence of oxygen. Same comment for paragraph 3.2.
- justify the choice of pure anatase TiO2 as it exhibits a lower catalytic activity than a mixture of anatase and rutile.
- clarify the text related to Figure 4a. Why does the yield in CO gradually decrease until 2EG-CuTi and then strongly increase for 3 and 5EG-CuTi ? and the opposite trend for the CH4 yield.
- the authors must compare the performance as well as the selectivity of the CuOx/TiO2 catalyst for CO2 reduction to other TiO2-based photocatalysts described in the literature and highlight the advances made.
- the quality of some figures could be improved. For example, the text inserted in Figures 5a-f can only hardly be seen.
- the language could be improved. The manuscript contains also a few typing errors.
Comments on the Quality of English LanguageMinor corrections are required
Author Response
The authors report the preparation of heterostructured photocatalysts associating anatase TiO2 and Cu(0) or copper oxides. The key role played by the oxidation state of Cu in the photocatalytic CO2 reduction was demonstrated. The work is of interest and the catalysts were well characterized. From my opinion, results could be better discussed in the context of literature. The following comments should be considered:
- introduction : "instability of Cu". Please clarify. Only Cu(0) and Cu+ are unstable in the presence of oxygen. Same comment for paragraph 3.2.
- Response: Thank you for your valuable suggestion, some references have been added in the revised manuscript..
The remarkable characteristic of Cu is the diversity of valence states (Cu2+, Cu+ and Cu0) and its instability. It is found that the valence state of Cu is unstable because copper and copper are easily oxidized in the presence of oxygen. Through a series of experiments and characterization methods, it is revealed that in the presence of oxygen, the oxidation reaction of mono-valent copper and zero-valent copper will affect its performance and stability in the photocatalytic reaction[2, 3].The role of different valence states of Cu in photocatalytic CO2 reduction has attracted more and more attention. - justify the choice of pure anatase TiO2 as it exhibits a lower catalytic activity than a mixture of anatase and rutile.
- Response: Thank you for your suggestion. The crystalline phase of TiO2 are anatase, rutile, plate titanite and a mixture of anatase and rutile (e.g. P25). Anatase phase titanium dioxide has a good basis for photocatalytic activity and is easy to be treated and modified. Surface modification and modification can be done by doping, loading and other methods to improve its photocatalytic activity and stability[9, 10]. More importantly, the crystal structure of anatase phase titanium dioxide is relatively simple, the property is more stable, and it is easy to be characterized and analyzed.
- clarify the text related to Figure 4a. Why does the yield in CO gradually decrease until 2EG-CuTi and then strongly increase for 3 and 5EG-CuTi ? and the opposite trend for the CH4 yield.
- Response: Thank you for your suggestion. The crystalline phase of TiO2 are anatase, rutile, plate titanite and a mixture of anatase and rutile (e.g. P25). Anatase phase titanium dioxide has a good basis for photocatalytic activity and is easy to be treated and modified. Surface modification and modification can be done by doping, loading and other methods to improve its photocatalytic activity and stability[9, 10]. More importantly, the crystal structure of anatase phase titanium dioxide is relatively simple, the property is more stable, and it is easy to be characterized and analyzed.
- the authors must compare the performance as well as the selectivity of the CuOx/TiO2 catalyst for CO2 reduction to other TiO2-based photocatalysts described in the literature and highlight the advances made.
- the quality of some figures could be improved. For example, the text inserted in Figures 5a-f can only hardly be seen.
- Response: Thank you for your suggestion. Figures 5a-f has been adjusted to increase the text size.
- the language could be improved. The manuscript contains also a few typing errors.
- Response: The language has been improved carefully.
Round 2
Reviewer 1 Report
Comments and Suggestions for Authors
I believe all comments have been thoroughly addressed and the manuscript is now ready for publication.
Author Response
I believe all comments have been thoroughly addressed and the manuscript is now ready for publication.
Thank you for your comments and suggestions.
Reviewer 2 Report
Comments and Suggestions for Authors
Few changes were made by the authors.
Some of my comments were also not considered.
Author Response
- Few changes were made by the authors. Some of my comments were also not considered.
- Response: Thank you for your valuable suggestion. Our statement was wrong and has been corrected.
The remarkable characteristic of Cu is the diversity of valence states (Cu2+, Cu+ and Cu0). Through a series of experiments and characterization methods, it is revealed that in the presence of oxygen, the oxidation of cuprous and zero-valent copper will affect its performance and stability in the photocatalytic reaction.The role of different valence states of Cu in photocatalytic CO2 reduction has attracted more and more attention.